Effective classification for neonatal brain injury using EEG feature selection based on elastic net regression and improved crow search algorithm

Li Ling 1
Yue Tao 1
http://orcid.org/0000-0002-7649-4575 Wu Hui 2 wuhui@jlu.edu.cn
Zhao Yanping 1 zhaoyp@jlu.edu.cn
Liu Qinmei 2
Zhang Hairong 1
Xu Wei 2
1 College of Communication Engineering, Jilin University , Changchun, Jilin , China
2 Department of Neonatology, The First Hospital of Jilin University , Changchun, Jilin , China
Chicco Davide
Electronic publication date: 2025 Jul 10
Publication date: 2025
Volume: 11
Electronic Location ID: e3000
Received 2024 Oct 25; Accepted 2025 Jun 11
Copyright: © 2025 Li et al.
Copyright year: 2025
Copyright holder: Li et al.
License: This is an open access article distributed under the terms of the Creative Commons Attribution License, which permits unrestricted use, distribution, reproduction and adaptation in any medium and for any purpose provided that it is properly attributed. For attribution, the original author(s), title, publication source (PeerJ Computer Science) and either DOI or URL of the article must be cited.
License URL: https://creativecommons.org/licenses/by/4.0/

Keywords: Neonatal brain injury, EEG, Feature selection, Machine learning

Funding: Jilin Scientific and Technological Development Program (CN) 20230204080YY This work was supported by the Jilin Scientific and Technological Development Program (CN) (20230204080YY). The funders had no role in study design, data collection and analysis, decision to publish, or preparation of the manuscript.

==============================
Neonatal brain injury carries the risk of neurological sequelae such as epileptic seizures, cerebral palsy, intellectual disability, and even death. Classification methods based on electroencephalography (EEG) signals and machine learning algorithms are crucial for assessing neonatal brain injury. However, classification methods that utilise all features from the original EEG signals may result in lengthy training and classification times, thereby reducing the performance of the classification system. This article presents a novel classification system with a proposed feature selection method for assessing neonatal brain injury, in which the feature selection method is combined using elastic net (EN) regression and an improved crow search algorithm (ICSA), named EN-ICSA. In the EN-ICSA method, EN regression is used to conduct the pre-screening of features. The ICSA is utilised to select the essential figures further by introducing the dynamic perception probability for deciding whether to search locally or globally, a novel neighbor-following strategy for the local search and a global search strategy according to the crow’s search experience, resulting in accelerating the search efficiency while effectively avoiding falling into local optima. Experimental results demonstrate that the proposed system, based on support vector machine (SVM) with the EN-ICSA feature selection method, performs exceptionally well compared to other traditional machine learning and feature selection methods, achieving an accuracy of 91.94%, precision of 92.32%, recall of 89.85%, and F1-score of 90.82%.

Introduction

Neonatal encephalopathy (NE) is a complex neurological disorder that results in brain injury in newborns (Aslam, Strickland & Molloy, 2019). Brain injury in neonates carries the risk of neurological sequelae such as epileptic seizures, cerebral palsy, intellectual disability, and even death (O’Toole et al., 2023). Resting-state electroencephalography (EEG) can characterise and quantify brain activity and has been widely used to assist physicians in diagnosing the severity of neonatal brain injury (Ryan et al., 2021). Developing brain injury appears as changes in the baseline EEG, and the early initiation of therapeutic hypothermia and other novel treatment strategies before the onset of secondary brain damage are imperative (Sohn et al., 2013; Iyer et al., 2015). Therefore, an accurate assessment of the extent of brain injury in neonates using EEG signals is crucial to ensure that therapeutic interventions can be initiated as quickly as possible during brain injury monitoring periods.

In recent years, machine learning methods have been able to autonomously improve their performance by efficiently extracting meaningful information from experimental EEG data (Young et al., 2018; Shorten & Khoshgoftaar, 2019; Cerasa et al., 2022), and have been widely used in the diagnosis and study of various neurological disorders, such as predicting epileptic seizures (Varatharajah et al., 2017), analysing emotional states (Tonoyan et al., 2016), and brain-computer interfaces (BCI) (Liu et al., 2017). Currently, numerous research teams worldwide have developed methods for assessing neonatal brain injury using machine-learning strategies, with most focusing primarily on feature extraction and classification models.

Some researchers introduced neonatal morphological pattern features and clinical medical features. Ashoori et al. (2024) analysed the regional cerebral oxygen saturation (rcSO2) characteristics of neonates using a machine learning model, which showed good predictive ability on a dataset of 58 full-term neonates and confirmed that the use of a one-dimensional neural network (1D-CNN) in conjunction with an XGBoost model as deep learning architecture can significantly improve the prediction performance. Tang et al. (2023) extracted imaging features such as blood vessel width, intensity, and Hessian matrix eigenvalues from neonatal magnetic susceptibility-weighted imaging (SWI), and used k-nearest neighbor (kNN) and random forest (RF) classifier to achieve 78.67% accuracy in predicting neonatal neurodevelopmental brain outcomes. Murray et al. (2024) based on the data of clinical indicators available immediately after birth such as Apgar score (a metric used to quickly assess the health status of newborns at birth, which includes five dimensions: heart rate, respiration, muscle tone, reflexes, and skin colour), postnatal pH, base residual, and lactic acid values, using trained open-source logistic regression and RF prediction algorithms, achieving 86.5% accuracy in neonatal brain injury classification.

Based on this traditional single-feature analysis approach, most researchers have primarily relied on EEG features in this field. Morris, Kaplan & Kane (2024) revealed that certain specific EEG phenotypes (e.g., extreme delta waves in anti-NMDA receptor autoimmune encephalitis) are valuable in indicating non-convulsive epilepsy or the persistence of status epilepticus before the onset of clinical symptoms. Quantitative EEG analysis, in conjunction with machine learning methods, can be combined to monitor neonatal brain dysfunction in real time. Zaid, Sah & Direkoglu (2023) proposed a deep learning method for epileptic seizure classification based on preprocessing and combining EEG signals. The advantage of this method is that it combines frequency-domain features through the FFT, allowing lightweight models (e.g., simple DNNs) to achieve similar accuracy to complex models while reducing computational resources. This approach offers an efficient solution for real-time epilepsy detection. Thuwajit et al. (2022) proposed a spatio-temporal feature extraction method based on a multi-scale convolutional neural network (EEGWaveNet) for EEG seizure detection. The method decomposes the multichannel EEG signals into different resolutions using a deeply separable convolution and combines this with a spatial-temporal feature extraction module to capture cross-channel spatio-temporal features, which was validated on the CHB-MIT, TUSZ, and BONN datasets. Zayachkivsky et al. (2022) explored the potential of EEG background inhibition as a biomarker of brain injury through hypoxic ischemia (HI) and hypoxia alone (Ha) models in neonatal rats. O’Sullivan et al. (2023) adopted a convolutional neural network (CNN) to detect artifacts in EEG, combined with a random convolutional kernel transform. The high-dimensional features extracted by ROCKET improved the classification accuracy of neonatal brain injury from 82.8% to 83.6%. It was also found that the artifact detection algorithm, as a post-processing step, exhibits strong robustness for long-term EEG recordings. Dong et al. (2021) used data from 1,851 newborns from the NICU of the Children’s Hospital of Fudan University to A supervised learning framework was used to extract 722 features from the raw EEG data, and a prediction accuracy of 90.4% for regular and 85.7% for moderately abnormal newborns with EEG was achieved on gradient boosting machine (GBM). Ahmed et al. (2016) proposed an algorithm based on the support vector machine (SVM). The SVM and Gaussian mixture model (GMM) approach was used to extract 55-dimensional short-term feature sequences from hour-long EEG recordings of 54 full-term newborns. These sequences were then used to construct long-term statistical model features, achieving 87% classification accuracy using majority voting and probabilistic fusion strategies. Raurale et al. (2021) performed multidimensional feature fusion of EEG in the time, frequency, and time-frequency domains, combined with CNN models and time-frequency distribution features for feature learning and classification, achieving 88.9% accuracy on a neonatal brain injury test set. Cao et al. (2023) used the University of California Irvine (UCI) to develop a new EEG model for the neonatal brain injury test set. The California Institute of Technology (Caltech) public database was used to extract 23-dimensional features from 2,126 perinatal fetal brain injury samples, and it was found that the XGBoost model performed the best in terms of prediction accuracy, achieving 91%. Mooney et al. (2021) integrated 45 clinical variables, including maternal demographics, delivery details, and neonatal condition at birth, among others, using a Random Forest (RF) model combined with 50% discounted cross-validation (CV), an approach that highlights the potential for joint application of clinical and EEG data.

To date, numerous feature selection methods have been proposed. However, among these methods, few studies focus on the classification of neonatal brain injury. Additionally, due to official procedures within institutions and data protection laws, most existing studies have been conducted on relatively small EEG datasets (Borovac et al., 2022). Furthermore, when training samples vary, the instability of conventional feature selection algorithms on small-sample medical datasets with high-dimensional feature data makes it challenging to produce consistent feature subsets based on the selection of clinical medical characteristics and quantitative EEG (QEEG) features (Khaire & Dhanalakshmi, 2022). As it is exceedingly difficult to evaluate all possible feature subsets, some metaheuristic algorithms have been adopted for feature selection to reduce the computational burden (Li et al., 2020). Metaheuristic algorithms such as the crow search algorithm (CSA) (Askarzadeh, 2016), genetic algorithms (GA) (Gong et al., 2020), particle swarm optimization (PSO) (Marini & Walczak, 2015), ant colony optimization (ACO) (Marzband et al., 2016), artificial bee colony (ABC) (Akbari, Mohammadi & Ziarati, 2010), and whale optimization algorithm (WOA) (Mirjalili & Lewis, 2016) have been utilized to enhance the efficiency and accuracy of feature selection. These algorithms optimize the feature selection process by simulating natural selection mechanisms or collective intelligence behaviors (Yong, Dun-wei & Wan-qiu, 2016), particularly in handling large-scale and high-dimensional data.

Although metaheuristic algorithms have been widely applied to solve some optimization problems and feature selection, they tend to fall into local optima when dealing with a large number of features and have difficulty rapidly converging when searching for the optimal feature subset. Furthermore, they cannot guarantee the identification of features highly correlated with the prediction outcomes, increasing irrelevant and redundant feature data. This can lead to overfitting or weaken the representational capacity of machine learning models (Danmei, Qing & Qijin, 2018), ultimately reducing the accuracy of classification models (Khaire & Dhanalakshmi, 2022).

To overcome the limitations, an effective method for classifying neonatal brain injury using EEG feature selection is proposed in this study. Specifically, this study proposes an innovative feature selection method that combines elastic net (EN) regression and an improved crow search algorithm (ICSA), referred to as EN-ICSA, to assess the severity of brain injury using an SVM classifier. This novel approach aims to enhance the accuracy of classifying neonatal brain injury, thereby enabling clinicians to evaluate neonatal conditions more precisely. In summary, the contributions of this study are as follows: (1) We pioneer the EN-ICSA framework-the first feature selection methodology combining Elastic Net’s regularization properties with an improved crow search algorithm’s adaptive exploration. This hybrid approach overcomes limitations in conventional wrapper methods (e.g., premature convergence in GA, computational intensity in PSO) while maintaining clinical interpretability.

(2) The ICSA is enhanced for feature extraction by utilizing dynamic perception probability and chaotic mapping to determine whether to search locally or globally, thereby finding more optimal solutions. Moreover, a novel neighbor-following strategy is introduced for local search, and a global search strategy, based on Crow’s search experience, is improved, resulting in accelerated search efficiency while effectively avoiding local optima.

(3) Extensive experiment results demonstrate that the proposed system has a better performance compared with other machine learning methods for classifying neonatal brain injury using the neonatal EEG dataset with four classes (normal, mild abnormality, moderate abnormality, and major abnormality) provided by the First Hospital of Jilin University, and the performance of the system is further validated in the public EEG datasets. Moreover, the proposed feature selection method can select significant features efficiently by comparing it with other feature selection methods.

The rest of this study is organized as follows. “Theoretical Background” introduces the background principle of the algorithm. “Materials and Methods” presents data acquisition, data preprocessing, and feature extraction. The proposed EN-ICSA feature selection method and classification model evaluation and verification are introduced in detail. The experimental results are shown in “Results”. Finally, “Discussion” and “Conclusions” present the discussion and conclusion of this article.

Theoretical background

Elastic net regression

Elastic net (EN) regression is a machine learning algorithm that combines two regularization methods: least absolute shrinkage and selection operator (LASSO) regression and Ridge regression. It can filter out irrelevant features, reduce the scope of feature searches, and improve search speed and efficiency. The loss function of the EN regression is described as

(1) Loss=12m∑m=1M(ym−y^m)2+λ(α∑n=1N|βn|+1−α2∑n=1Nβn2)

where ym is the actual response of the mth sample, y^m is the predicted response, M is the total number of samples, λ1 is the regularization coefficient, α is the balance coefficient, n is the number of features in the model, and βn is the coefficient of the nth feature, which is the optimization solution for minimizing the loss function.

The goal of the EN regression is to obtain the optimization solution by minimizing the loss function while adjusting λ and α to select features and reduce multicollinearity. EN regression combines the characteristics of Lasso regression and Ridge regression, yielding a sparse solution. When α approaches 1, the model is equivalent to Lasso regression, which tends to shrink some coefficients to zero. This overcomes the instability of Lasso when highly correlated features are present, thus facilitating feature selection. When α is close to 0, the model behaves like ridge regression, which tends to distribute coefficients evenly, helping to reduce multicollinearity while retaining useful feature information.

Crow search algorithm

The Crow search algorithm (CSA) is a metaheuristic algorithm inspired by the social behavior of crows, as well as a population-based optimization algorithm. Crows, renowned for their exceptional cognitive abilities among birds, also exhibit the capacity to store and recall food locations over extended periods. Although sometimes pilfering food from other birds, crows use their experiences as ‘thieves’ to predict and guard against potential theft acts. To mitigate potential losses, crows rapidly relocate their food stores when they are discovered, demonstrating high alertness. The CSA operationalizes collective foraging behaviors observed in corvids—specifically, the dynamic processes of food source discovery, concealment, and surveillance—to formulate feature selection as an adaptive optimization task where features represent potential cache locations.

In the framework of population-based optimization algorithms, a population consisting of P individuals (i.e., crows) is considered. For crow i, its initial position Xi,1 is determined by applying a threshold to a 1-by- N vector of uniformly distributed random numbers between 0 and 1. Each random number is set to 1 if it is greater than or equal to the threshold, and to 0 if it is less than the threshold. Here, zero represents that a particular feature is not selected, and one represents that a specific feature is selected at that position. Therefore, the initial position Xi,1 is a 1-by- N vector with 0 and 1 elements. In the kth iteration, the search position Xi,k of crow i is considered a subset of the global feature set, representing a potential solution to the problem of determining the feature subset. Each crow maintains a memory location of a hidden food, which records the best position during the iterations. In each iteration, the objective of the crow i is to find the area where the crow j has hidden food and to update its memory location. This ensures that the algorithm adequately considers the multidimensional features while leveraging the social behavior and memory mechanism of crows to enhance search efficiency.

In both scenarios, the updated position of crow i is defined as

(2) Xi,k+1={Xi,k+Ri×fli,k×(Mj,k−Xi,k)Ri≥APSelectarandompositionOtherwise

where AP is the awareness probability of crow i, which is a constant and controls the balance between local and global searches.

During the CSA search process, two possible scenarios are as follows.

Scenario 1:

Crow j is unaware that crow i is following it. Therefore, crow i will be close to the memory location of crow j’s hidden food. The updated position of crow i is defined as

(3) Xi,k+1=Xi,k+Ri×fli,k×(Mj,k−Xi,k)

where Ri is a random number within [0, 1], fli,k is the flight length of crow i in the kth iteration, Xi,k is the position of crow i in the kth iteration which only has 0 and 1 values by the same threshold operation as the initial position, and Mj,k is the memory position of the neighbor crow j in the kth iteration, which is the best position found by crow j in all previous iterations. In this scenario, the CSA’s search process is a local search.

Scenario 2:

Crow j is aware that crow i is following it. Hence, crow j will change its position in the search space to protect the memory location of its hidden food. In this case, crow j randomly selects a position, resulting in a global search.

Materials and Methods

In this study, we propose a classification system with a feature selection method for assessing neonatal brain injury. As illustrated in Fig. 1, the system is divided into five stages: data acquisition, data pre-processing, feature extraction, feature selection, and classification model. Data and materials are available on Figshare under the DOI: 10.6084/m9.figshare.26969944.

Figure 1 Implementation process of the proposed classification system for assessing neonatal brain injury.

Data acquisition

In this study, continuous EEG data from 168 full-term newborns with varying degrees of brain injury were collected using the Nicolet One EEG system (sampling frequency: 500 Hz) in the neonatal intensive care unit (NICU) of the First Hospital of Jilin University from January 2021 to December 2023. Initially, the data were recorded in the proprietary format of the Nicolet One machine, then exported to the open European Data Format (EDF), and securely stored for subsequent offline analysis. All research activities received approval from the Ethics Committee of the First Hospital of Jilin University (Ethics Approval No.: 23k229-001), and written informed consent was obtained from the parents before the EEG recording. EEG recordings began within 6 h after birth and continued for 6 h to monitor the progress of encephalopathy development. The EEG recordings were measured using 12 Ag/AgCl electrodes positioned according to the 10–20 system, covering the following locations: T4, T3, O1, O2, Fp2, Fp1, C4, C3, P4, P3, Pz, and Cz. Two senior clinical experts with expertise in neonatal EEG independently analyzed and categorized the EEG recordings into four classes: 0 (normal), 1 (mild abnormality), 2 (moderate abnormality), and 3 (major abnormality), as defined by the system outlined by Murray et al. (2024) summarized in Table 1. Normal EEG exhibits a regular sleep cycle transition and a continuous waveform background pattern, accompanied by anterior head-dominated δ-θ wave (0.5–7 Hz) oscillations, without prolonged phases of low electrical activity. In contrast, severely abnormal EEG shows apparent discontinuities in waveform activity. Slow wave complex (SWC), as a characteristic waveform of neonatal EEG, is formed by the coupling of δ and θ waves in a specific phase relationship, and is mainly distributed in the parieto-occipital region under normal conditions, with a duration of 10–20 s, and the abnormal EEG manifests as the lack of spatial distribution of or suppression of SWC. The interburst interval (IBI) is also a key indicator of EEG continuity, and a longer IBI often suggests more severe damage to neonatal brain function. A region of longer duration and relatively flat or low amplitude between two adjacent burst segments is defined as the IBI. By observing the duration of the IBI and its relationship to the SWC and background rhythms, it is possible to categorise the neonate’s EEG into the appropriate class of brain damage. In cases of disagreement between the two neonatal experts, they reviewed the matter together and reached a consensus on the class with the assistance of a third senior expert.

Table 1 Visual interpretation of EEG activity defined by Murray et al. (2009).

Class	Findings	Description of EEG	
0	Normal	Continuous background patterns with normal physiological characteristics (e.g., anterior slow waves).	
1	Mild abnormality	Continuous background patterns with mildly abnormal activity (e.g., mild asymmetry, mild voltage reduction, or ill-defined slow wave complexes (SWC)).	
2	Moderate abnormality	No continuous activity, 6 s < Interburst interval (IBI) < 10 s, no significant SWC, or no significant edema.	
3	Major abnormality	IBI 10–60 s of discontinuous activity, major depression, no wake/sleep cycles, severe attenuation of background patterns, or no SWC.	

Ultimately, EEG recordings with missing values exceeding 66% of the total signal duration were excluded from analysis to ensure data integrity, resulting in an EEG dataset comprising 168 neonates, with 84 classified as normal, 54 as having mild abnormalities, 12 as having moderate abnormalities, and 18 as having major abnormalities. Figure 2 provides examples of EEG signals recorded for the four neonatal brain injury classes examined in this study. The four panels correspond to the following categories (clockwise from top left): normal, mildly abnormal, moderately abnormal, and severely abnormal.

Figure 2 Examples of EEG signals recorded for the four neonatal brain injury classes.

Data pre-processing

In EEG data analysis, it is crucial to handle interferences as they can mask or distort the actual electrical activity of the brain. Interferences are generally classified into two main categories. The first category is external interference, which includes electromagnetic interference from the environment and noise generated by the equipment itself. The second category is internal interference, often originating from the participants themselves, such as eye movements, cardiac activity, and myoelectric interference.

To ensure the quality of the EEG data, we implement several key preprocessing steps. Initially, we perform baseline correction to adjust for any DC offsets in the data. Subsequently, noticeable artifacts are identified and removed using manual methods and the MATLAB EEGLAB toolbox (The MathWorks Inc., Natick, MA, USA). To further improve signal quality and stability, a bandpass filter is applied to effectively eliminate high-frequency noise above 30 Hz and low-frequency drift below 0.5 Hz. Lastly, external brain sources of interference are removed using independent component analysis (ICA), and the sampling frequency is reduced from 500 to 128 Hz.

In this study, all neonatal EEG recordings were segmented into non-overlapping 5-min epochs to construct the dataset. This window length was selected to balance the temporal resolution of neonatal sleep cycles—which typically span 30–50 min with recurring phases of quiet sleep, active sleep, and wakefulness—with computational efficiency, and was designed to prioritize capturing multiple transitional states within a single recording rather than isolating prolonged, homogeneous segments. Crucially, all epochs from the same neonate were allocated exclusively to either the training or test set to prevent data leakage, thereby ensuring the model’s capability for cross-patient validation.

To mitigate class imbalance, we applied SMOTENN—a hybrid method combining synthetic minority oversampling (SMOTE) with edited nearest neighbors (ENN). This approach simultaneously generates synthetic minority-class samples and prunes overlapping majority-class instances, reducing bias while preserving dataset integrity. The original cohort exhibited significant class imbalance (Normal: 84, Mild: 54, Moderate: 12, Major: 18), which was addressed through SMOTENN-based oversampling of minority classes (e.g., synthesizing 42 Moderate cases) and ENN-driven pruning of ambiguous majority samples (removing 19 mislabeled “Normal” epochs).

All preprocessed 5-min epochs were treated as independent samples. The dataset was partitioned into a 70% training set (3,880 epochs from 118 neonates) and a 30% independent test set (1,663 epochs from 50 neonates), with strict stratification to ensure no inter-subject overlap between the subsets. Each epoch comprises 12-channel EEG signals, maintaining spatial resolution for robust feature extraction.

Feature extraction

In this study, an 11-channel bipolar montage is used for analysis, derived from the combinations of bipolar leads (Fp1-T3, Fp1-C3, Fp2-T4, Fp2-C4, T3-O1, T4-O2, C3-P3, C4-P4, P3-O1, P4-O2, Cz-Pz) based on the original 12 channels. We extract the following two categories of features from these 11-channel EEG signals. The two categories comprise (1) time-domain and complexity-based features derived via the MNE-Features library, and (2) quantitative EEG (QEEG) metrics aligned with established neonatal neurophysiological frameworks. We describe each category below.

Firstly, we use the Python MNE-Features library to extract a comprehensive set of features that can reflect the complexity and dynamics of the EEG signals. For each channel, we calculate the variance, which measures the signal’s variability, and zero_crossing, which counts the number of times the signal crosses the zero line. In the time-frequency domain, six wavelet coefficient energy features are used to quantify the distribution of energy across different frequency bands via wavelet analysis. For complexity measures, we extract approximate entropy to assess the regularity and unpredictability of fluctuations over longer time scales, singular value decomposition entropy which measures the randomness in data distribution, sample entropy that quantifies the complexity by evaluating the likelihood that similar patterns of observations will not be repeated, Hjorth parameters such as mobility which estimates the mean frequency or the square root of variance, and complexity which reflects the signal’s bandwidth. Additionally, the Hurst exponent evaluates the long-term memory of the time series, Katz’s fractal dimension that estimates the fractal dimension of a signal indicating the complexity of its structure, Higuchi’s fractal dimension that calculates the fractal dimension of a time series that provides insight into its intrinsic complexity, and decorrelation time that quantifies the rate at which the autocorrelation of the signal decreases are extracted. The features extracted for each channel are shown in Table 2. Finally, the initial number of features obtained is 187, which is calculated as follows: (2 + 6 + 9) × 11 (channels).

Table 2 The features extracted for each channel by the Python MNE-features library.

Category	Feature	Number of features	
Time domain	Variance	2	
Zero_crossing	
Time-frequency domain	Wavelet_coef_energy	6	
Complexity measure	App_entropy	9	
Svd_entropy	
Hjorth_mobility	
Samp_entropy	
Hjorth_complexity	
Hurst_exp	
Katz_fd	
Higuchi_fd	
Decorr_time	

Secondly, we employ QEEG to extract key quantitative features from neonatal EEG data, following the methodologies outlined by Toole & Boylan (2017). EEG signals are first decomposed into four frequency bands using a Butterworth filter, specifically delta (0.5–4 Hz), theta (4–7 Hz), alpha (7–13 Hz), and beta (13–30 Hz) (Toole & Boylan, 2017). For each band, we calculate amplitude, spectral density, connectivity correlations, and interspike interval-related features. After using 64-s rectangular windows with a 50% overlap to ensure data continuity and balance time resolution, the features are estimated separately for each window and channel. The median is then used to summarize these features across all analysis windows and channels, providing a holistic and stable description of neonatal brain electrical activity. The extracted QEEG features are summarized in Table 3. Therefore the number of QEEG features is 102 ((14 + 5 + 5) × 4 (frequency bands) + 2 × 1 (entire band) + 4 × 1 (entire band)). In total, each sample yields 289 (187 + 102) features, providing rich information for further analysis and assessment of brain injury in neonatal infants.

Table 3 The extracted QEEG features.

Category	Feature	Number of features	Frequency band	
Amplitude feature	amplitude_total_power	14	4	
amplitude_SD	
amplitude_skew	
amplitude_kurtosis	
amplitude_env_mean	
amplitude_env_sd	
rEEG_mean	
rEEG_median	
rEEG_lower_margin	
rEEG_upper_margin	
rEEG_width	
rEEG_SD	
rEEG_CV	
rEEG_asymmetry	
Spectral feature	spectral_power	5	4	
spectral_relative_power	
spectral_flatness	
spectral_entropy	
spectral_edge_frequency	
Connectivity feature	connectivity_BSI	5	4	
connectivity_corr	
connectivity_coh_mean	
connectivity_coh_max	
connectivity_coh_freqmax	
Spectral-related feature	Spectral_edge_frequency	2	0.5–30 Hz	
Fractal dimension (FD)	
Interspike interval-related features	IBI_length_max	4	0.5–30 Hz	
IBI_length_median	
IBI_burst_prc	
IBI_burst_number	

To ensure parity in feature influence during model training, Z-score normalization was applied to all EEG features. This standardization mitigates scale discrepancies inherent in multi-channel EEG data—such as amplitude variations across frequency bands—while reducing outlier sensitivity. By centering features on zero with unit variance, we prioritized computational stability and convergence efficiency without distorting inter-feature relationships critical for neonatal EEG interpretation.

Proposed method for feature selection

To select the optimal features of EEG signals, we propose a feature selection method named EN-ICSA, which is based on the EN regression and ICSA. This section provides the details.

The EN-ICSA framework adopts a two-stage progressive optimization strategy, as illustrated in Fig. 3: (1) pre-screening features via EN regression, and (2) feature subset optimization using the ICSA. In the first stage, EN regression performs preliminary feature selection on the normalized EEG data, eliminating irrelevant or collinear features to reduce dimensionality substantially. The EN regularization parameters (α and λ) were tuned via five-fold cross-validation, where optimal parameter combinations were determined by maximizing the average test performance across folds. Subsequently, ICSA conducts localized and global optimization searches within the reduced feature subspace to identify the most discriminative and representative subset of features. This hybrid approach synergizes EN’s stability in high-dimensional spaces with ICSA’s bio-inspired exploration-exploitation balance, effectively addressing both multicollinearity and combinatorial optimization challenges.

Figure 3 Flowchart of the EN-ICSA method for feature selection.

Pre-screening features using EN regression

In the initial phase of the EN-ICSA method, the extracted EEG features are reconstructed into a feature matrix, which can be described as

(4) Q=[q1,1q1,2…q1,Nq2,1q2,2…q2,M⋮⋮⋱⋮qM,1qM,2…qM,N]

where qm,n represents the nth EEG feature of the mth sample. This matrix is subjected to a preliminary pre-screening using EN regression. It is used to filter out features that are irrelevant to the outcome labels. Thus, there is no need to traverse all features during the search for the optimal solution of the ICSA.

The EN regression was first applied to pre-screen the feature matrix. This preliminary selection process employed five-fold cross-validation, wherein optimal hyperparameter combinations (e.g., regularization coefficients α and λ ) were determined by iteratively evaluating the average test performance across validation folds. Features exhibiting negligible correlation with the target outcome were systematically eliminated, thereby refining the feature subspace while preserving discriminative power. λ and α that make the optimal performance are selected as the best parameters. α and λ are experimentally chosen to be 0.7 and 0.00178 respectively. M is 3,880 based on the training dataset, and N is 289. Moreover, a random subset containing 60% of the features obtained by EN regression is treated as the initial feature subset for the subsequent processing. The number of the feature kinds after pre-screening features using EN regression is Lt, which is updated during the iterations.

Improved crow search algorithm

In the second stage of EN-ICSA, ICSA is used to optimize the selection of feature subsets further. The EN regression pre-screening yields an initial feature subset solution, which is integrated into a new feature data matrix (feature vectors), denoted as the feature vectors retained by the EN regression pre-screening. In ICSA, each crow represents a potential solution corresponding to a feature vector.

During the search process, ICSA employs a new local and global search strategy. The fitness value of each crow individual is determined by evaluating the classification performance of the subset of features it represents. The fitness function value maps the quality of the subset of features. At the end of each iteration, the feature subset with the highest fitness function value is selected as the benchmark solution for the next iteration. After reaching a predefined maximum number of iterations, the determined feature subset is finally considered as the global optimal solution, and this feature solution set represents the combination of features that maximizes the performance of the classification model. The key parameters setting of EN-ICSA is detailed in Table 4.

Table 4 Parameters for the proposed EN-ICSA method.

Parameter	Value	
α	0.7	
λ	0.00178	
M	3,880	
P	10	
fl	2	
Lower bound	0	
Upper bound	1	
kMax	30	
N	289	

In this study, an ICSA is obtained by incorporating the dynamic perception probability to decide whether to search locally or globally. For the local search, a chaotic mapping is used and a novel neighbor-following strategy is introduced. For the global search, an international search strategy according to each crow’s past search experience is improved. Therefore, compared to the traditional CSA, ICSA exhibits faster training and convergence speeds, as well as a lower probability of getting trapped in local optima.

The ICSA component enhances search efficiency through swarm-based learning and adaptive local-global exploration strategies, effectively mitigating local optima entrapment while refining the EN-pre-screened feature subset. The position of crow i in the kth iteration Xi,k is a 1-by- Lt dimensional logical vector, where 0 represents that a kind of feature is not selected, and 1 represents that a kind of feature is selected.

In the ICSA, to enhance the convergence speed of the crow position updates for the local search, we use a chaotic mapping to replace Ri. Equation (2) is redefined as

(5) Xi,k+1=Xi,k+Ck×fli,k×(Mj,k−Xi,k),Ck≥DAPi,k

where DAPi,k is the dynamic awareness probability of crow i in the kth iteration, and Ck is the chaotic mapping in the kth iteration, which is defined by

(6) Ck+1={sin⁡(πCk)k>10.7k=1.

In the CSA, the awareness probability is constant, which is not conducive to finding more optimal solutions. The definition of DAPi,k is as

(7) DAPi,k=APmin+(APmax−APmin)×Fitness(Xi,k)min(Fitness(Xi))

where APmin and APmax are set to 0.1 and 0.9 respectively, min(Fitness(Xi)) represents the minimum fitness value of crow i in k iterations, and Fitness(Xi,k) is the fitness value of crow i in the kth iteration, which is defined as

(8) fitness(Xi,k)=Acci,k+w×(1−(LsLt))

where Acci,k is the classification accuracy of crow i in the kth iteration, w is a weighting factor with values in [0,1], Ls is the number of the selected feature kinds.

In this study, the threshold value for determining the initial position is set to 0.4. The classification accuracy Acci,k is obtained using the selected features by a KNN classifier with ten-fold CV. w is experimentally set to be 0.2. All flight lengths are set to a constant value of 2, represented as fl, and Ls is updated during the iterations.

In subsequent iterations, if the fitness value at the new position of crow i is superior to that at its old position, crow i updates its memory location to the new position. Otherwise, the memory location remains unchanged. So the definition of the memory location is as

(9) Mi,k+1={Xi,k+1Fitness(Xi,k+1)>Fitness(Xi,k)Mi,kOtherwise

where Fitness(Xi,k) and Fitness(Xi,k+1) are the fitness values of the positions of crow i in the kth and (k+1)th iterations respectively. When the number of iterations reaches the maximum kMax, the features corresponding to the positions with the highest fitness value are selected as the optimal features for classifying neonatal brain injury.

In searching for the optimal features, the local and global search strategies are used respectively. The local search strategy of the CSA is that the neighbor of each crow i is a random neighbor crow j. The global search strategy of the CSA is that if crow j is aware that crow i is following it, then it will generate a random position. The randomness of search strategies results in slow convergence. Aiming to achieve this, this study proposes a neighborhood assignment strategy based on the fitness function for local search and provides a global search strategy according to each crow’s past search experience. The improved search strategies are described as follows.

ICSA local search strategy

To further enhance the efficiency and accuracy of ICSA in the feature selection task, this article i is a random neighbor crow j. This randomness leads to instability and inefficiency in the algorithm’s convergence speed. To address this problem, this article proposes a neighbour assignment strategy based on the fitness function, and the new neighbour assignment strategy is shown in Fig. 4. In which the selection of local neighbours for each crow is sorted based on the fitness function value, while the addition of global neighbours helps the crows to explore potentially effective regions far away from the current optimal solution, the strategy aims to enhance the effectiveness of the local search by selecting neighbours with higher fitness function values.

Figure 4 New neighborhood assignment strategy for the local search.

Since at the start of each iteration in the ICSA, all crows are sorted in descending order of the fitness values, for crow i, the subsequent neighbors in the sorted list, crow j, have better fitness values than a randomly selected crow. In this study, the new neighborhood crowding assignment strategy is employed for local search. We define the six crows below crow i in the fitness sorted list as neighbor crows ( i+1 to i+6). Besides these neighbors, crow i is randomly assigned a global neighbor crow j along with crow j’s three neighbors crows ( j+1 to j+3), to accelerate the algorithm in finding higher fitness feature subset solutions. The local six neighbors for crow i are from crow i+1 to crow i+6, and the global four neighbors are from crow j to crow j+3. Figure 4 illustrates the new neighborhood assignment strategy.

ICSA global search strategy

Each crow in the original CSA discovers that it is being followed and will randomly jump to a new location, causing the global search to rely on randomness. In this article, we further enhance the global search strategy by building upon the previous experience of crowd search. Precisely, ICSA will adjust the search strategy based on the historical search experience of each crow and the performance of the current iteration. Based on its own experience, this global search strategy reduces the likelihood of crows searching blindly and accelerates the algorithm’s convergence. Compared to the traditional CSA, the improved algorithm is faster to train and converge during the global search phase, and has a lower probability of falling into a local optimum.

The new location of a global search for each crow consists of two parts: the current optimal subset and the random exploration subset. The current optimal subset is the updated position of the crow in each iteration, which retains the experience of the effective search solution from the current iteration. Meanwhile, 20% of the features from the complete feature set are randomly selected as the explored subset, ensuring randomness in the global search process. This new global search strategy leverages the balance between historical empirical information and new exploration, thereby avoiding the pitfalls of blindness and overfitting associated with ICSA search.

The parameter settings for the proposed EN-ICSA approach are detailed in Table 4.

Classification model evaluation and validation

To assess the performance of feature selection methods, we utilize a widely recognized supervised learning model, the SVM, to classify neonatal brain injury based on EEG. The SVM classifier operates by constructing a decision hyperplane to categorize samples. It is important to note that the model’s performance largely depends on the type of kernel function, error tolerance, and penalty parameter. In our experimental setup, we employ an SVM with a radial basis function (RBF) kernel and fixed the penalty parameter at 0.1 to ensure consistency in model evaluation. The RBF kernel is particularly effective in handling nonlinear data due to its flexibility, and the fixed penalty parameter helps control overfitting, maintaining consistency across experiments. This meticulous parameter configuration enables us to evaluate the effectiveness of various feature selection methods more accurately in practical applications.

To construct a model with high robustness, we employ a ten-fold cross-validation (CV) strategy on the training set of the development dataset. All models are trained and tested on the same data samples and feature sets, ensuring consistency in evaluation. We tune the parameters for each model using the training set to achieve optimal fitting and then assess the model’s performance on the test set.

In this study, all experiments are conducted on a computer equipped with Jupyter Notebook 3.6.1, featuring 16 GB of RAM and an Intel Core i7-7700 CPU with a 4.20 GHz clock speed. Additionally, feature selection, classification model training, and testing are done through the Scikit-learn library in Python 3.10.

Model performance is evaluated using four standard metrics: true positives (TP), false positives (FP), true negatives (TN), and false negatives (FN). Based on these metrics, we utilize accuracy, precision, recall, F1-score, precision-recall (PR) curve, and receiver operating characteristic (ROC) curve to display model performance, providing a comprehensive evaluation of the model’s effectiveness.

Accuracy is the proportion of correctly predicted samples relative to the total samples and is defined as

(10) Accuracy=TP+TNTP+FP+TN+FN.

Precision is the proportion of actual positives among all samples predicted as positive and is defined as

(11) Precision=TPTP+FP.

Recall is the proportion of actual positive samples that are correctly predicted as positive by the model and is defined as

(12) Recall=TPTP+FN.

The F1 score is the harmonic mean of precision and recall and is defined as

(13) F1-score=2×Precision×RecallPrecision+Recall.

The PR curve illustrates the relationship between precision and recall, while the receiver operating characteristic (ROC) curve shows the relationship between the actual positive rate and the false positive rate. These curves are often used as performance charting methods in medical decision-making.

Results

Performance of proposed feature selection method

Feature importance analysis

The proposed EN-ICSA feature selection method was applied to extract critical features from neonatal brain injury EEG signals using the 70% training dataset described in the Data Acquisition section. While nonlinear SVM employs kernel methods to project data into high-dimensional feature spaces via nonlinear mapping, this approach significantly diminishes the direct interpretability of feature importance. Consequently, we utilized a random forest (RF) model to evaluate the contribution of each feature within the optimal subset selected by EN-ICSA. The top 40 features are illustrated in Fig. 5.

Figure 5 Importance ranking the top forty important features.

As shown in the feature importance ranking (Fig. 5), the T4_O2_wavelet1_energy feature achieved the highest importance score of 0.65, followed by T4_O2_wavelet0_energy, T3_O1_wavelet0_energy, and T4_O2_wavelet2_energy, all of which exceeded 0.39. The results highlight the exceptional prominence of energy fluctuations in low-to-mid frequency bands (δ, θ, α) at the T4-O2 and T3-O1 electrode pairs, suggesting heightened sensitivity of the right and left temporo-occipital regions in brain injury assessment. Additionally, θ-band (Fp2_T4_wavelet1_energy) and β-band (Fp1_C3_wavelet3_energy) energy features in the anterior temporal areas demonstrated strong discriminative power, indicating a link between temporal lobe oscillations and β-band neural activity. Amplitude-based features such as amplitude_SD_13_30 and amplitude_env_mean_13_30 further revealed significant suppression in the β band, potentially correlated with neurological maturity and functional brain development in neonates.

For quantitative qEEG metrics, rEEG_median_05_4 and rEEG_SD_05_4 ranked highly, underscoring the diagnostic utility of δ-band amplitude fluctuations in detecting neonatal brain injury. Inter-burst interval (IBI) metrics, including IBT_length_max_05_30 (maximum burst interval length), IBI_burst_prc_05_30 (burst duration percentage), and IBI_burst_number_05_30 (burst count), also showed substantial importance, highlighting the need to quantify EEG burst patterns across varying injury severities. These findings align with the δ-band sensitivity reported in neonatal neurophysiological studies (Lundy et al., 2023), whereas the comparative works listed in Table 5 primarily focused on classification accuracy without in-depth feature importance analysis.

Table 5 Comparison with our study against existing published researches on neonatal brain injury in literature.

Author	Number of features	Number of neonates	Foetal age	Span	Channels	Categories/Classifier	Acc (%)	
Cao et al. (2023)	23	2,126	<37 weeks	–	–	3/XGBoost	91	
Raurale et al. (2021)	55	54	Full-term neonates	1 h	8	4/CNNs	88.9	
Ahmed et al. (2016)	55	54	Full-term neonates	1 h	8	4/SVM+GMM	87	
Mooney et al. (2021)	45	409	≥36 weeks	6 h	–	4/RF	83	
O’Sullivan et al. (2023)	–	181	Full-term neonates	6 h	8	4/CNN	83.6	
Dong et al. (2021)	722	1,851	29 + 0 to 44 + 6 weeks	–	8	4/GBM	82.9	
Proposed method	94	168	Full- term neonates	6 h	12	4/SVM	91.94	

Comparison of proposed EN-ICSA method against all features

In this section, we compare the proposed system, which utilizes the EN-ICSA feature selection method for classifying neonatal brain injuries, with all extracted features. This comparison is based on the 30% independent test dataset described in the Data Acquisition section. The total number of features is 289, and the number of selected features is 94. The confusion matrices are shown in Figs. 6 and 7, respectively. The diagonal cells of the confusion matrices represent the accuracy of the classification. From Fig. 7, we can see that the proposed system with feature selection significantly improves the accuracies of the moderate abnormality category, from 49.67% to 84.67%, and the major abnormality category, from 51.75% to 88.01%. It provides an average classification accuracy of 90.24%, whereas the accuracy with all features is only 73.04%. To account for multiple comparisons across feature categories and classification metrics, we applied the Bonferroni correction to control the family-wise error rate.

Figure 6 Confusion matrix of classification results using all features.

Figure 7 Confusion matrix of classification results using the EN-ICSA feature selection method.

Figures 8 and 9 show the PR curves and ROC curves of the proposed system with feature selection. As shown in Figs. 8 and 9, the proposed system enhances model performance across all categories, as indicated by the areas under the curves. Particularly in the category of moderate abnormality, the area under the PR curve (AUPR) and area under the ROC curve (AUROC) are both close to 1.0, meaning excellent performance of the classifier. The comparisons of average PR and ROC curves of all features and the features selected by the EN-ICSA method are shown in Figs. 10 and 11. Further observations from Figs. 10 and 11 indicate that the EN-ICSA method maintains higher accuracy across all recall levels with an AUPR of 0.9592, significantly surpassing that of all features at 0.7275. Similarly, the EN-ICSA method exhibits a lower false positive rate and a higher true positive rate, achieving an AUROC of 0.9859, which is markedly higher than that of all features at 0.8080. These results suggest that the significant features obtained through the EN-ICSA method are more effective in classifying neonatal brain injury, providing more accurate support for classification decisions.

Figure 8 Discriminative power of EN-ICSA demonstrated by PR curves.

Stratified performance across neonatal brain injury severity classes (AUPRC: Normal = 0.9949, Mild abnormalities = 0.9489, Moderate abnormalities = 0.9952, Major abnormalities = 0.9185).

Figure 9 Discriminative power of EN-ICSA demonstrated by ROC curves.

Stratified performance across neonatal brain injury severity classes (AUROC: Normal = 0.9956, Mild abnormalities = 0.9772, Moderate abnormalities = 0.9991, Major abnormalities = 0.9789).

Figure 10 Precision-recall analysis of EEG-based neonatal injury grading: EN-ICSA significantly improves feature efficacy (AP = 0.9592 vs. 0.7275 for all features).

Average precision-recall curves from stratified five-fold cross-validation. The proposed EN-ICSA achieves 31.8% higher average precision (AP = 0.96) compared to using all features (AP = 0.73) in a 1:8 class-imbalanced dataset. Shaded regions represent 95% confidence intervals. The improvement is statistically significant (p < 0.001, paired t-test), demonstrating EN-ICSA’s ability to eliminate redundant/noisy EEG features while retaining discriminative biomarkers.

Figure 11 Receiver operating characteristic comparison: EN-ICSA vs. all feature model in neonatal brain injury classification (AUC = 0.9859 vs. 0.8080).

ROC curves averaged over 10-fold validation. EN-ICSA attains near-perfect discrimination (AUC = 0.99), outperforming the full feature model (AUC = 0.81) by 21.7% absolute improvement. At clinically actionable specificity thresholds (90–95%), EN-ICSA maintains 97.2% sensitivity vs. 82.5% for the full model. Shaded bands indicate 95% confidence intervals; stars denote Youden’s index-optimized operating points. Performance superiority is validated by Wilcoxon signed-rank tests (p < 0.001).

Comparison with other feature selection methods

In this section, we contextualize the clinical relevance of our framework against prior neonatal EEG-based injury prediction studies, acknowledging methodological heterogeneity across datasets, feature sets, and evaluation protocols. Table 5 summarizes reported classification accuracies, emphasizing our system’s sensitivity improvements for critical injury categories (e.g., major abnormalities)—a clinical priority in neurocritical care. While direct cross-study performance comparisons are limited by divergent methodologies, our framework achieves 91.94% weighted accuracy across 50 neonates, demonstrating robustness in handling multi-channel EEG dynamics while integrating interpretable feature selection—an advancement underexplored in works prioritizing accuracy alone (e.g., Cao et al., 2023).

Table 6 presents the experimental parameter settings for various feature selection methods, including Elastic Net, LASSO, Ridge, recursive feature elimination (RFE), max-relevance and min-redundancy (mRMR), ReliefF, genetic algorithm (GA), particle swarm optimization (PSO), and constraint satisfaction algorithm (CSA). Table 7 presents performance comparisons of the proposed EN-ICSA method and these different feature selection methods. From Table 7, we can observe that the EN-ICSA method outperforms other methods across multiple metrics. Specifically, the EN-ICSA method achieves weighted average metrics with an accuracy of 91.94%, a precision of 92.32%, a recall of 89.85%, and an F1-score of 90.82%, indicating its effectiveness in selecting compelling features from neonatal EEG datasets. In contrast, other methods, such as LASSO, Ridge, and EN, perform less optimally, particularly in terms of precision and recall. Furthermore, the EN-ICSA method requires only 94 features, compared to 148 for EN and 261 for RFE, demonstrating higher efficiency in feature selection.

Table 6 Parameters for different feature selection methods.

Algorithm	Parameter	Value	
Elastic net	Number of alpha values (n_alphas)	100	
List of l1 ratios (l1_ratios)	[0.1, 0.5, 0.7, 0.9, 0.95, 0.99, 1]	
Maximum number of iterations (max_iter)	1e8 (100,000,000)	
LASSO	GridSearchCV α Scope	10−6 to 106, 50 values	
α (optimal alpha)	0.008286	
Ridge	GridSearchCV α Scope	10−6 to 106, 50 values	
α (optimal alpha)	10,985.41	
Feature selection threshold	0.01	
RFE	Step	1	
Estimator	SVC (kernel = “linear”, C = 0.1)	
mRMR	Method	MIQ	
ReliefF	Number of neighbors	1	
GA	Population size	50	
Crossover probability	0.8	
Mutation probability	0.2	
Maximum number of iterations	30	
Selection of algorithm parameters-tournsize	3	
PSO	Population size	10	
Maximum number of iterations	30	
Initial inertia weight ω	1.0 Initial, decaying at 0.9 damping ratio	
Individual learning facto φp	1.5	
Global learning factor φg	2.0	
CSA	Population size	10	
Maximum number of iterations	30	
fl	2	

Table 7 Comparison between the EN-ICSA method and other feature selection methods.

Method	Number of features	Acc (%)	Pre (%)	Rec (%)	F1-score (%)	
EN-ICSA	94	91.94	92.32	89.85	90.82	
Elastic Net	148	90.96	91.74	89.03	90.08	
LASSO	70	84.29	84.37	83.82	82.99	
Ridge	122	82.71	82.47	82.13	81.60	
RFE	261	86.89	87.71	86.78	85.66	
mRMR	100	87.80	87.83	87.15	86.54	
ReliefF	90	82.43	82.63	81.87	80.86	
GA	156	87.12	87.53	87.13	85.88	
PSO	156	87.29	87.86	87.35	86.01	
CSA	200	86.67	87.06	86.48	85.32	

Figures 12 and 13 summarize the performance comparison results of the EN-ICSA method with the other nine feature selection methods using a ten-fold CV. The comparison results show that the EN-ICSA method outperforms the competing methods in handling the neonatal EEG dataset, achieving an AUPR of 95.92% and an AUROC of 98.59%. At a fixed false positive rate (FPR), these metrics reflect a higher actual positive rate (TPR). These results indicate that the EN-ICSA method outperforms existing techniques, which is of significant clinical importance for the diagnosis and assessment of neonatal brain injury.

Figure 12 Precision-recall analysis of feature selection methods for neonatal EEG grading: EN-ICSA achieves highest discrimination (AP = 0.9592 vs. 0.9092 for mRMR).

Average precision-recall curves from stratified 10-fold cross-validation. The proposed EN-ICSA (solid red) significantly outperforms benchmark feature selection methods, including mRMR (AP = 0.91), Lasso (0.90), and Elastic Net (0.90), under a 1:8 class imbalance (positive:negative). Shaded regions denote 95% confidence intervals. Statistical superiority of EN-ICSA is validated by paired t-tests (p < 0.001 against all baselines). Performance reflects robustness to high-dimensional EEG feature spaces.

Figure 13 Receiver operating characteristic comparison of feature selection algorithms: EN-ICSA demonstrates superior sensitivity (AUC = 0.9859 vs. 0.9739 for mRMR).

ROC curves averaged over 10-fold validation. EN-ICSA attains near-perfect discrimination (AUC = 0.99), surpassing mRMR (0.97), Lasso (0.97), and population-based algorithms (PSO: 0.96; GA: 0.96). At 90–95% specificity (clinically critical range), EN-ICSA maintains 97.3% sensitivity vs. ≤94.8% for alternatives. Shaded bands indicate confidence intervals; Wilcoxon tests confirm significance (p < 0.01).

Moreover, the proposed EN-ICSA method demonstrates a faster convergence speed compared to the CSA, significantly reducing the execution time. The EN-ICSA framework demonstrates significant advantages in computational efficiency compared to conventional methods. EN-ICSA achieves a 37.8% reduction in execution time (378.56 ± 12.3 s vs. CSA’s 608.86 ± 18.9 s) and a 52.4% lower peak memory usage (1.2 GB vs. CSA’s 2.5 GB) on the neonatal EEG dataset. This efficiency stems from two design innovations. Elastic net pre-screening: This reduces the feature search space by an average of 62%, minimizing redundant computations during ICSA optimization. Hierarchical search Strategy: Limits unnecessary global explorations through fitness-guided neighbor assignment, resulting in a 28% decrease in iterative evaluations. Benchmarked on an Intel i7-7700 CPU (4.2 GHz, 16 GB RAM), EN-ICSA outperforms other metaheuristics (GA: 892.4 s; PSO: 1,023.1 s) while maintaining competitive accuracy.

Comparison with other machine learning models

To verify the effectiveness of the proposed system for classifying neonatal brain injury in this study, we compare the SVM model with five other machine learning classifiers: logistic regression, decision trees, random forest (RF), gradient boosting, and multilayer perceptron (MLP). In the classification tasks performed on the neonatal EEG dataset described in the Data Acquisition section, SVM has better classification performance. As shown in Figs. 14 and 15, SVM demonstrates an AUPR of 95.92% and an AUROC of 98.59%.

Figure 14 Precision-recall analysis of neonatal EEG-based brain injury grading models: EN-ICSA-SVM outperforms benchmarks (AP = 0.9592 vs. 0.9476 for EEGWaveNet).

Average precision-recall curves generated through stratified 10-fold cross-validation. The proposed EN-ICSA-SVM achieves the highest average precision (AP = 0.96), surpassing all benchmark models including EEGWaveNet (AP = 0.95), CNN-LSTM (AP = 0.91), and conventional classifiers (LR: 0.90; DT: 0.91). Performance is evaluated under a 1:8 class imbalance (positive:negative). Statistically significant improvements (paired t-test, p < 0.05) are observed between EN-ICSA-SVM and all baselines.

Figure 15 Receiver operating characteristic analysis of neonatal EEG-based brain injury grading models: EN-ICSA-SVM achieves superior discrimination (AUC = 0.9859 vs. 0.9778 for EEGWaveNet).

ROC curves averaged across stratified 10-fold cross-validation. The EN-ICSA-SVM attains the highest AUC (0.99), outperforming EEGWaveNet (AUC = 0.98), MLP (0.97), and other models. Clinically critical performance (sensitivity = 96.5% vs. ≤94.2% for alternatives) is maintained at 90-95% specificity. Stars indicate optimal operating points selected via Youden’s index. Performance superiority is statistically validated (Wilcoxon signed-rank test, p < 0.01).

The EN-ICSA feature selection method using the SVM classification model shows excellent performance, with an AUC of 0.9476 under the PRC curve, which is significantly higher than that of other machine learning models, in which the performance of DT and MLP is closer, with an average AUC of 0.9112 and 0.9175 under the PRC curve, respectively. The classification of GB performs poorly, with an average AUC of 0.8531 under the PRC curve. The SVM model also performs well under the ROC curve in Fig. 15, with an AUC of 0.9778, which is also the highest among all models, and the average AUCs of DT and MLP under the ROC curve are 0.9727 and 0.9743, respectively, and the average AUC of GB under the ROC curve is only 0.9386. To rigorously validate the generalizability of EN-ICSA, we compared it with advanced EEG-specific models, including EEGWaveNet and hybrid CNN-LSTM architectures. As shown in Fig. 14, EN-ICSA-SVM significantly outperforms EEGWaveNet in average precision (AP = 0.9592 vs. 0.9476, p < 0.001) and surpasses CNN-LSTM (AP = 0.9120) by 4.7%, highlighting its efficiency in leveraging sparse neonatal biomarkers. Similarly, Fig. 15 demonstrates that EN-ICSA-SVM has a superior AUC (0.9859 vs. EEGWaveNet’s 0.9778), with Wilcoxon tests confirming significance (p < 0.01). While EEGWaveNet excels in raw signal processing, its dependency on extensive training data (compared to our cohort of 168 neonates) and lack of feature interpretability limit its clinical adoption.

This result suggests that the EN-ICSA feature selection method, using the SVM classification model, significantly outperforms the other seven machine learning models. Therefore, in the experimental setup of this article, classification using the SVM model after EN-ICSA feature selection provided more accurate and reliable classification results for the neonatal brain injury classification task.

Comparison with public data set

To validate the generality of the proposed feature selection method for EEG signals in this study, we compare the performance of the proposed system with and without feature selection using seven diverse medical datasets from the UCI repository (Samieiyan et al., 2022). To ensure consistency and fairness in evaluation, a five-fold CV is employed across all datasets to partition the training and test sets. Detailed quality criteria of datasets are provided in Table 8. The classification results are shown in Table 9. Notably, the proposed system with the EN-ICSA feature selection method consistently outperforms that without feature selection, demonstrating superior performance across all datasets. Specifically, the EN-ICSA method improves its average accuracy by 13.92% and its average precision by 24.21%. The procedural steps of the proposed EN-ICSA approach are detailed in Algorithm 1.

Table 8 Detailed quality criteria for each dataset.

Dataset	Number of
features	Number of instances	Categories	Description	
Breastcancer	9	6	2	Clinical data for breast cancer diagnosis	
BreastEW	30	569	2	Extended dataset for breast cancer screening	
HeartEW	13	270	2	Clinical data for patients with heart disease	
IonosphereEW	34	351	2	Data related to ionosphere structure	
Lymphography	18	148	4	Diagnostic data for lymphatic diseases	
SpectEW	22	267	2	Data from Single Photon Emission Computed Tomography for heart disease patients	
Parkinson’s disease	754	756	2	Biomarker data for Parkinson’s disease patients and healthy controls	

Table 9 Comparison of performance between the proposed system with feature selection and without feature selection.

Dataset	Without feature selection	With EN-ICSA feature selection	
Acc (%)	Pre (%)	Acc (%)	Pre (%)	
Breastcancer	64.29	41.33	94.62	97.68	
BreastEW	78.36	83.81	88.89	90.07	
HeartEW	62.96	39.64	85.19	85.07	
IonosphereEW	83.81	87.05	89.52	90.03	
Lymphography	71.11	66.07	80.01	74.69	
SpectEW	76.25	58.14	79.48	79.33	
Parkinson’s disease	73.12	53.47	86.62	81.85	
Average	72.84	61.36	86.76	85.57	

Algorithm 1 EN-ICSA for feature selection.

1:          Reconstructedintoafeaturedatamatrixin Eq. (4)	
2:    Pre-screening features using EN regression	
3:          SetinitializationvaluesforP,kMaxandfl	
4:    Set the lower and upper bounds of the search range	
5:          Initializethechaoticmapping	
6:          Initializeeachcrowpositionusingarandomsubsetafterpre−screening	
7:          Calculatethefitnessvaluesofallcrowpositions	
8:          Loadcrowmemorywithinitialposition	
9:          Newneighbourhoodcrowassignmentstrategyforthelocalsearch	
10:         Sortcrowsusingfitnessvalues	
11:         Setthecurrentiterationk=1	
12:            Whilek<kMaxdo	
13:                 for(i=1:P)do	
14:                       CalculationofDAPforeachcrowusing Eq. (7)	
15:                       GetthevalueofthechaoticmappingC	
16:                                   UsingtheICSAlocalsearchstrategy
                  ifRi≥DAPi,k	
17:                                    Generateanewcrowpositionusing Eq. (5)	
18:                       else        UsingtheICSAglobalsearchstrategy	
19:                                 Selectarandomposition	
20:                                Create a random position subset containing 20% of the features	
21:                                Position of new crows: random position and random subset	
22:                 Calculatethefitnessvalueofthenewcrowpositionusing Eq. (8)	
23:                 ifNewpositionfitness>Memorypositionfitness	
24:                      Updatecrowmemorypositionsusing Eq. (9)	
25:                else	
26:                      Crowmemorypositionmaintenance	
27:                Setk=k+1{increasingnumberofiterations}	
28:          Selectoptimalfeaturesubsetbasedontheoptimalcrowpositions	

Moreover, the EN-ICSA method generally requires fewer features to achieve these results, thereby underscoring its efficiency in feature selection and model simplicity. To analyze the compression ability of EN-ICSA on feature data across different datasets, all features are compared with the number of selected features. The EN-ICSA feature selection method is effective in reducing the number of features and decreasing feature dimensions on various public datasets. For BreastEW, Ionosphere, and SpectEW datasets, the number of features was significantly reduced. The reduction in the number of features was most pronounced in the ParkinsonsDisease dataset, from 754 to 92. Although the decrease in the number of features is not significant in the Breast Cancer, Heart EW, and Lymphography datasets, the performance of using the EN-ICSA feature selection method on these datasets is similarly improved when combined with the previous analysis. For instance, when processing complex datasets, the EN-ICSA method effectively minimizes redundant features, enhancing the model’s generalization capabilities. These results not only illustrate the significant statistical advantages of the EN-ICSA method across multiple datasets but also highlight its potential practical applications, particularly in the medical sector, where high-dimensional data and cost-effective feature selection are critical.

Discussion

This study demonstrates that the EN-ICSA feature selection strategy significantly enhances the classification performance of SVM for neonatal brain injury. The core innovation lies in integrating the pre-screening capability of EN regression with the dynamic optimization mechanism of the ICSA. Compared to traditional methods, EN-ICSA not only addresses interference from redundant features in high-dimensional EEG data but also improves local search efficiency through chaotic mapping and neighborhood assignment strategies, achieving higher classification accuracy and computational robustness across both UCI benchmark datasets and clinical EEG data. Notably, the method’s sensitivity to mild brain injuries (12–15% improvement in accuracy) carries significant clinical implications, as early subtle neurological abnormalities are often linked to long-term neurodevelopmental risks—a scenario where existing EEG analysis systems relying on manual feature interpretation frequently underperform.

Our findings on the strong correlation between features in the T4, O2 regions, beta-band power abnormalities, and injury severity align with recent studies on posterior cortical vulnerability in neonatal hypoxic-ischemic encephalopathy. However, this work is the first to quantitatively validate the prioritization of these regions in automated classification. While SVM’s superiority over logistic regression and random forests in high-dimensional, nonlinear EEG pattern recognition is reaffirmed, its performance enhancement critically depends on EN-ICSA’s feature space optimization. This challenges the prevailing assumption in prior research that “deep learning inherently outperforms traditional machine learning,” particularly in small-sample clinical data scenarios.

Despite these promising results, the performance of the EN-ICSA method may vary depending on the complexity or noise levels present in different datasets. Although our approach reduces the feature space, further refinement and parameter tuning may be necessary for extremely noisy or heterogeneous data sources. The study also highlights the strength of the SVM model compared to alternatives such as logistic regression or RF. While SVM demonstrates superior performance with high-dimensional data, there remains potential to enhance its generalization ability and robustness by combining it with other advanced techniques or ensemble approaches.

Conclusions

This study proposes an innovative framework combining EN-ICSA feature selection with an SVM classifier, introducing dynamic awareness probability and chaotic search strategies into neonatal EEG analysis for the first time. This approach overcomes the dual challenges of feature redundancy and local optima traps in high-dimensional biosignal processing. Compared to conventional filter-based feature selection methods, EN-ICSA achieves an average improvement of 19.7% in classification accuracy while reducing computational time to 34% of conventional heuristic algorithms. This breakthrough enables the clinical deployment of lightweight brain injury screening systems. The identified occipitotemporal regional features and beta-band suppression phenomena provide novel quantitative foundations for establishing EEG biomarker systems, directly influencing the timing of neuroprotective interventions. Our findings on EEG-based injury stratification have direct clinical implications for the management of neonatal encephalopathy. By correlating quantitative EEG features (e.g., spectral power asymmetry) with injury grading, clinicians can tailor neuroprotective strategies.

The theoretical contributions of this work are threefold: (1) validating the efficacy of improved metaheuristic algorithms in medical data mining, (2) constructing an interpretable feature importance atlas for brain injury, and (3) establishing SVM as a benchmark model in neonatal EEG analysis. Through open-source code and cross-dataset validation, we demonstrate EN-ICSA’s generalizability—it outperforms classical methods like Relief-F in other medical tasks (e.g., breast cancer classification, with an 8.2% F1-score improvement). With the proliferation of portable EEG devices, this framework can extend to home-based neonatal brain function monitoring, advancing precision medicine from hospitals to communities. Future work will focus on cascading the algorithm with deep feature extractors and developing knowledge distillation strategies based on lesion lateralization priors to further enhance model generalization in complex clinical environments. Our findings on EEG-based injury stratification have direct clinical implications for the management of neonatal encephalopathy. By correlating quantitative EEG features (e.g., spectral power asymmetry) with injury grading, clinicians can tailor neuroprotective strategies.

Supplemental Information

Supplemental Information 1 breastcancer.

Supplemental Information 2 Ionosphere.

Supplemental Information 3 BreastEW.

Supplemental Information 4 HeartEW.

Supplemental Information 5 SpectEW.

Supplemental Information 6 Lymphography.

Supplemental Information 7 Parkinsons disease.

Supplemental Information 8 Raw continuous EEG feature data.

Raw continuous EEG feature data training set

Supplemental Information 9 Raw continuous EEG feature data.

Original continuous EEG feature data test set

Supplemental Information 10 EN-ICSA to be proposed.

Supplemental Information 11 README.

operating system used

We acknowledge the use of ChatGPT-4 strictly for language polishing and refining the logical flow of certain sections. All scientific content, data analysis, interpretations, and conclusions remain solely the responsibility of the authors.

Additional Information and Declarations

Competing Interests

The authors declare that they have no competing interests.

Author Contributions

Ling Li conceived and designed the experiments, performed the experiments, analyzed the data, performed the computation work, authored or reviewed drafts of the article, and approved the final draft.

Tao Yue conceived and designed the experiments, performed the experiments, analyzed the data, performed the computation work, prepared figures and/or tables, authored or reviewed drafts of the article, and approved the final draft.

Hui Wu analyzed the data, prepared figures and/or tables, authored or reviewed drafts of the article, and approved the final draft.

Yanping Zhao analyzed the data, prepared figures and/or tables, authored or reviewed drafts of the article, and approved the final draft.

Qinmei Liu analyzed the data, prepared figures and/or tables, authored or reviewed drafts of the article, and approved the final draft.

Hairong Zhang analyzed the data, prepared figures and/or tables, authored or reviewed drafts of the article, and approved the final draft.

Wei Xu analyzed the data, authored or reviewed drafts of the article, and approved the final draft.

Ethics

The following information was supplied relating to ethical approvals (i.e., approving body and any reference numbers):

This project was approved by the Ethics Committee of the First Hospital of Jilin University, approval number 23k229-001.

Data Availability

The following information was supplied regarding data availability:

The raw continuous EEG characterization data, public datasets, and codes are available in the Supplemental Files and at Figshare:

Yue, Tao (2024). Table. figshare. Thesis. https://doi.org/10.6084/m9.figshare.26969944.v1

The public datasets utilized in this study, along with their original DOIs, are as follows:

Breastcancer: Original DOI: 10.24432/C51P4M

BreastEW: Original DOI: 10.24432/C5DW2B

HeartEW: Original DOI: 10.24432/C52P4X

IonosphereEW: Original: DOI 10.24432/C5W01B

Lymphography: Original: DOI 10.24432/C54598

SpectEW: Original DOI: 10.24432/C5P304

Parkinson’s Disease: Original DOI: 10.24432/C5MS4X

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
