# Peer review of "Effective classification for neonatal brain injury using EEG feature selection based on elastic net regression and improved crow search algorithm"

_PeerJ Computer Science, doi:10.7717/peerj-cs.3000_

## Round 0.1 · original submission · Major Revisions

The article cannot be accepted in its current form. The reviewers raised a number of issues that need to be addressed by the authors. Please prepare a new version addressing these issues.

Reviewer 1 ·

Basic reporting

The introduction lacks sufficient comparison with state-of-the-art (SOTA) works like EEGWaveNet and other advanced EEG-based seizure detection frameworks. These are highly relevant due to the shared challenges of EEG signal analysis in neonatal brain injury and seizure detection. Integrating these comparisons will better position the novelty of EN-ICSA.

Language is generally clear but could benefit from professional editing to refine grammar and avoid overly complex sentences, especially in the introduction and methodology sections.

Figures and tables need more explicit captions. For example, explain key metrics or highlight differences between EN-ICSA and alternative methods directly in the captions.

Statistical validation of results (e.g., p-values) is missing and should be included to substantiate claims of superiority over other methods.

Experimental design

Justify the assumption that seizure detection algorithms (e.g., EEGWaveNet) are transferable to neonatal brain injury classification. Provide evidence or rationale for why the underlying EEG signal features (e.g., frequency or amplitude changes) are comparable.

To ensure reproducibility, include more implementation details for EN-ICSA, such as specific hyperparameter values, data preprocessing steps, and computational resource requirements.

Discuss the small dataset size (168 neonates) and class imbalance (e.g., only 12 moderate abnormality cases). Address how this might affect model generalizability and performance.

The dataset split (70%/30%) is reasonable, but further validation using cross-dataset or external testing is recommended to strengthen claims of generalizability.

Validity of the findings

Include statistical tests (e.g., paired t-tests, ANOVA) to demonstrate that performance improvements achieved by EN-ICSA are significant compared to baseline and SOTA methods.

Provide a more detailed analysis of misclassifications. For example, why does EN-ICSA show reduced accuracy in certain classes (e.g., mild abnormalities)?
Extend comparisons to include other advanced EEG feature selection or classification algorithms (e.g., EEGWaveNet, hybrid deep learning-metaheuristic models) to demonstrate EN-ICSA's broader applicability.

The discussion should emphasize clinical relevance, specifically how EN-ICSA’s computational efficiency and accuracy improvements can impact real-time neonatal brain monitoring.

Additional comments

Consider expanding the literature review to include SOTA works like EEGWaveNet and recent EEG-based classification methods to strengthen the context of your work.

Figures illustrating EEG patterns across different brain injury classes would provide valuable insight into the signal characteristics influencing model performance.

Discuss potential extensions of EN-ICSA, such as integrating it with deep learning frameworks for end-to-end learning inspired by EEGWaveNet and other multiscale architectures.

Address ethical considerations, particularly regarding data sharing and clinical applicability. Ensure compliance with privacy standards if this system is to be adopted for real-world use.

Reviewer 3 ·

Basic reporting

In this study, the authors present a novel classification system for neonatal brain injury using EEG feature selection. The proposed approach integrates Elastic Net (EN) regression and an Improved Crow Search Algorithm (ICSA) to optimize feature selection, thereby improving classification accuracy. The study uses EEG data from 168 full-term newborns collected at the First Hospital of Jilin University and evaluates the performance of the proposed method against traditional feature selection techniques. The results indicate that the EN-ICSA method achieves superior accuracy (91.94%) and efficiency compared to other approaches. The study contributes to the field by introducing a feature selection method tailored for high-dimensional neonatal EEG data. After thoroughly reviewing the manuscript, I have some concerns and suggest minor revisions before acceptance. Please see my specific comments below.
• The study uses SVM with an RBF kernel but does not justify why SVM was chosen over deep learning methods (e.g., CNNs, LSTMs).
• The manuscript describes the feature selection process well but lacks details on hyperparameter tuning for the SVM model. How were the regularization parameter (C) and kernel parameters (gamma) optimized? Was a grid search or cross-validation approach used?
• The study identifies T4, O2, and beta-band suppression as important EEG markers for neonatal brain injury. It would be helpful to discuss how these findings align with clinical knowledge.
• Did the statistical analyses (e.g., correlation between FA and disease duration) undergo Bonferroni correction or any other method to control for multiple comparisons? If not, the authors should clarify whether the results remain significant after such corrections.
• Please provide information on the age at onset for neonates with brain injury in the dataset.
• If the Jankovic Rating Scale (JRS) was used for classification, please describe the scoring criteria in more detail.
• Since some neonatal brain injuries affect specific brain regions asymmetrically, did the study consider lateralization effects in feature selection?
• Figure captions need to be elaborated
I suggest a minor revision to improve clarity and strengthen the manuscript by addressing these concerns. Once these issues are addressed, the paper will be a valuable contribution to neonatal brain injury classification using EEG.

Experimental design

nothing to add

Validity of the findings

nothing to add

Reviewer 4 ·

Basic reporting

This paper proposed a combination method of Elastic Net Regression and Crow Search Algorithm for neonatal brain injury classification using EEG signal. The study provided a detailed methodology, experimentation, and relevant literature references. However, several areas require major revision to enhance clarity, reproducibility, and scientific rigor.

1. In term of English presentation, grammatical issues and phrasing ambiguities should be addressed for better readability.

2. The novelty of the ICSA modifications should be better articulated—how does this differ significantly from standard CSA

3. Although the paper mentioned some related work in the introduction section, it is encouraged to have a Related Work section to show the relevant work in this research field, the research gaps, and how your work is different this relevant work.

Experimental design

1. The study aligns with the journal's aims and presents a clear research question.

2. The background paragraph from Section 2 (MATERIALS AND METHODS) should be separated and present before Section 2.

3. Please explain the data preprocessing steps. Why do you divide your data set to a ratio 70/30 from training and testing? What about validating?

4. Please provide the details of how you label your dataset?

5. How sensitive is the EN-ICSA method to parameter changes? Does performance degrade significantly with different settings?

Validity of the findings

1. The study claims that EN-ICSA reduces computational resource compared to the other methods, but only a small paragraph on execution time back up for this claim.

2. Are there biases in class distributions?

3. How do you deal with the Class Imbalances?

4. How does the proposed method compare with clinician assessments in terms of classification reliability?

5. The study identifies T4, O2, Fp1, Fp2, C3, and C4 as important EEG channels for neonatal injury classification. However, why these channels are important from a clinical perspective should be further discussed

---

## Round 0.2 · Minor Revisions

The reviewers found a number of aspects that need to be improved. Please prepare a new enhanced version of the manuscript by taking into account the suggestions from the reviewers.

·

Basic reporting

The submitted manuscript is well written in proper English, however, there's a slight overuse of definite articles ("the"), although it is bearable. But it would be nice while refining the submission to work on this particular issue.

While line 192 is understandable, it seems to be a bit overly complex statement just to have a "flag" set in order to determine the use of features or not; maybe rephrase. Also, I would suggest to not write "The random number" instead I would prefer either "A random number", "Random numbers", or "Each random number".

Several type settings and other layout problems are visible:
line 199/200 crow j has a line break in between, maybe use a non-breaking space character.

Type setting issue in line 435 (italic fonts all the way through), type setting 443 indices I and j are not italic.

Although transfer can be made between paragraphs where crow j is the "lead" while crow I follows (199- 218) and paragraphs where crow I has the memory locations (426 to 439), it would be preferable to be consistent within this document to have an index for the crow that has the food hidden (memory location) and the crow that follows. Currently, readers need to focus really hard in the subsequent paragraphs where you describe the "neighbor"-assigning method. Here, the manuscript would benefit and improve readability if you'd stick to specific indices for each variation.

All in all this is complaining on a high level. Good job.

Experimental design

The experimental setup and design is convincing. Expertise in both - computer science and engineering, as well as medical experts - is given, which is highly recommended for conducting studies with bio-signal classification. The input data (EEG) looks mostly artifact free. It is also described in their data pre-processing section. Most of the times (especially with other public datasets) there are ECG interferences hidden beneath the signal, which seems completely free of it (figure 2).
Typical processing steps (utilizing MNE-Features Library) is done.
There are mutual exclusive metrics used.

All the steps above lead to the conclusion of a valid and nicely designed study on classification using EEG features, selected by the new method.

However, in the comparisons later on there's some EEGNet appearing (figure 14 and 15) which hasn't been introduced before. Please be so kind to name this specific method within your text as you refer to it as something very important ("surpassing all benchmark models including EEGNet").

Parameters are listed in order to reproduce experiments conducted.

Validity of the findings

Overall findings seem promising and valid.
There is a comparison of feature sets from other (publicly available) data sets. It would be nice to at least get some more information about the features being responsible for the EN-ICSA in outperforming the standard approach.

·

Basic reporting

The “Introduction” session provides adequate context and makes clear what the motivation is. However, it is not clear why the paper starts by introducing Neonatal Encephalopathy (NE), suggesting in this way that these words and this specific topic are crucially relevant throughout the paper, but then never discusses it again.

Literature is well referenced, paraphrased, and relevant, and is presented in the introduction session.

In the introduction, you anticipate 3 main contributions. However, it seems that they are just a summarization of your work. As “contribution”, I would instead expect you stressed the novelties of your study compared to others. So, for example, you should underline whether this is the first study applying SVM to this type or task, and that you developed, applied, and validated a new features selection method, contributing to existing methods. Results should not be reported at this stage and rather included in the conclusions to further stress the contributions of your work.
I noted some minor comments (note that row numbers refer to the reviewing pdf):
- The word “Ridge” is sometimes written with capital letters, sometimes not. Check for consistency;
- In Fig. 1 you wrote “artefact” (British English) instead of “artifact” (US), used in the rest of the article;
- The sentence in rows 190-191 is not clear and what you try to explain is confusing;
- Rows 209-210 suffer from incorrect english (“.. is the position of crow in the iteration with only has 0 and 1 values by the same threshold operation as the initial position”). Maybe there is a typo and you meant “which” instead of “with”?
- In row 217 you wrote “In both scenarios…” and then provided definition 3. However, since the definition indeed regards both scenarios, it would have been better to define it once at the beginning, rearranging the discussion of the algorithm;
- In row 254 you wrote “consensus score”, which is not written anywhere else in the paper. However, it is not clear what "score" refers to after you added in this revised version the description of the 4 classes. Saying “consensus on the class” would make it clear;
- In row 255 you wrote: “more than 66% of EEG recordings were excluded from analysis”. Could you briefly explain why, just like you did in the response letter?
- Which are the 4 classes in Figure 2? Text is too small and impossible to read, therefore you should provide the information on which picture corresponds to which class in the Figure description;
- In the formulas presentations, clarity may be enhanced by specifying the range of subscripts where applicable, e.g., “with i=1,...,n, where n is …”.

Experimental design

The article content is within Aims and Scope of the journal and article type, and the investigation meets ethical standards and technical quality.
Regarding the model selection methods, it is not clear if you attempted to use other classification models and eventually chose SVM for its superior performance, or if the comparison with other models (from line 609) came after you a priori chose SVM, without justifying this choice. If the former, change the implementation description saying that you tried this approach and finally chose SVM. Otherwise, explicitly say why you preferred SVM (for example, in the response letter you report “Furthermore, SVM’s decision boundaries provide clinicians with transparent criteria (e.g., T4-O2 δ-energy thresholds), crucial for actionable diagnostics in NICUs.”). You indeed addressed model selection in the conclusions, but this should be done earlier, whereas it is not useful to discuss it in the final section.
Discussion on data preprocessing is present, required and sufficient. However, you should integrate more details on class rations after balancing. On the other hand, your text is affected by several repetitions and a lack of concision, particularly in the discussion of the feature selection algorithm. The abundance of redundant information significantly affects clarity and readability, resulting in unnecessarily long text which confounds the reader. I would suggest revising this part to make content more concise and, thus, precise.
Below, I report some notes in relation to specific sections.

In the pre-processing section,
- I would add a reference to the two main categories of interferences discussed;
- In rows 275-279: “This window length was selected to balance neonatal sleep cycle characteristics with computational efficiency, designed to maximize the inclusion of neonatal EEG data rather than prioritizing fewer samples with extended observation periods”. Which are the characteristics of neonatal sleep cycle relevant to the window length choice? Adding a specification would help understanding the reasoning behind the choice, as well as the meaning of “to maximize the inclusion of neonatal EEG data”, which appears as a generic statement. Moreover, I believe the English form of the sentence is not clear. I would adjust it as follows: “.. efficiency, to maximize the inclusion…”.
- In rows 280-287, the description of the oversampling algorithm is rather long and does not integrate useful information that you provided in the response letter regarding the concrete effect of applying this method in your analysis, rather than the general description you provided in the revised version. Please, integrate the information on class ratios after balancing;


In the “Feature Extraction” section,
- There is no need to specify the features names for the first set of features. Moreover, after you wrote “We extract the following two categories of features from these 11-channel EEG signals. ”, to enhance clarity of the text, I would anticipate shortly which are the two categories, and then proceed with the description you provided afterwards;
- The period of sentence between 303-309 is quite long, I would rather include a count (for example by using “(i), (ii)”, etc.) or dots/semicolons to improve readability. Regarding the second features set, Table 3 solves any doubt, but you could make the discussion clearer by specifying the number of e.g. amplitude features (14), spectral density features, etc., as the final count of the number of features would otherwise result hard to comprehend (without the support of Table 3);
- Rows 330-335 are very repetitive as you write twice that you apply feature normalization. They are abundant in pros of this operation in a general context, not tailored to your context. You could make this part more concise.

In the “Proposed Method for Feature Selection” section,
- remove rows 337-343 and leave only 344-345, which summarize them. In rows 347-348 there is no need of specifying again what the EN-ICSA acronyms stand for.

In the “Pre-screening features using EN regression” section ,
- in row 360, i would remove “making it suitable for processing by the feature selection method. This matrix” and substitute it with “which”;
- notice that the part you now added is written using the past tense, whereas the rest is written in the present form: consistency is required.

In the “Improved crow search algorithm” section,
- “The key parameter settings of EN-ICSA are detailed in Table 4.” should be “The key parameterS setting of EN-ICSA IS detailed in Table 4.”;
- At row 398, you wrote “In the EN-ICSA feature selection method, the ICSA is used for further feature selection to derive the optimal subset of features after pre-screening features using EN regression.” You are repeating a concept you highly exposed elsewhere, and even within the same paragraph (see the beginning of this very paragraph), making the text poorly clear and confusing, as meaningful information gets lost in this repetitive concept;
- in row 408 “... iteration, AND Ck is the…”;
- Definitions from row 412 to row 419 are a unique long period which suffers from lack of clarity. Try to separate it into at least two periods.

In “Classification Model Evaluation and Validation” section,
- part of the information reported here, referred to the tools used for preprocessing and features extraction, was already present in the corresponding sections, making its repetition here unnecessary and confusing;
- you define the SVM acronym, but you already did it earlier, so it is not needed to do it again.

Validity of the findings

- How do findings exposed in the features importance analysis section relate to literature evidence? For example, do studies reported in Table 5 perform feature importance analysis or not? Provide adequate references to relevant studies;
- in rows 557-558 you wrote “As shown in Figs. 8 and 9 demonstrate that the proposed system enhances model performance across all categories..” is not correct in english;
- it is reported that the number of retained features based on the proposed method is 94, but it would have been interesting to have this information earlier, and not just in relation to a comparison with model performance using all the features.
- statistical tests should be reported not only in Figures caption but also in the main text. Moreover, in presenting results, it should be made explicit which are the hypotheses being tested and why you chose a specific test, in a statistically rigorous way;
- it is unclear how the Table 5 “comparison of the proposed method with those used by other researchers” is executed. Did you simply compare the accuracy scores of studies not sharing dataset, features selection method, features, classifier and number of classes? Do you think this is a robust methodology? Moreover, discussing this comparison in the section dedicated to the comparison with other features selection methods makes it confusing what the section is about;
- you refer to Figure 14 and 15 which compare SVM against more than 5 models, as you wrote when referring to the Figures. The other 2 models are discussed later, but you should anticipate the presence of their comparison, otherwise looking at the picture would suggest incongruence with respect to what you initially declared in the text;
- in row 618 you write “MLP” but this acronym was not defined anywhere earlier;
- “Shaded regions represent 95% confidence intervals” in Figure 14, but there are no shaded regions in the figure;
- for results in Table 9, why did you choose only accuracy and precision as performance metrics in this case? I would add a note on this in the text;
- why “The procedural steps of the proposed EN-ICSA approach are detailed in Table 10” only before the discussion section and after you have extensively discussed the algorithm in earlier sections? This should be moved earlier in the text;
- please integrate the Bonferroni correction implementation in the paper (which you exposed in the response letter).


The opening part of the Discussion section currently functions more like a summary of your work and would be more appropriately placed in the Conclusion, as it reiterates what the study is about rather than interpreting the findings. Regarding the form, consider making it more concise and focused specifically on interpreting the results in light of existing literature and highlighting the novel contributions of your work. For example, the misclassification comments you provided in the response letter could be integrated in this part, as well as the comment on how “EN-ICSA’s computational efficiency and accuracy improvements can impact real-time neonatal brain monitoring” (as suggested by reviewer 1-comment 12) which you included instead in the results section.
Regarding the conclusion section, its content overlaps considerably with the Discussion section, except for the final paragraph on future developments and features choice. Please, clearly separate the last two sections: in the Discussions,interpret and contextualize your findings, and reserve the Conclusion for assessing the impact and novelty of your work.

Additional comments

While I consider this study methodologically valid and of meaningful contribution to the domain, exposition suffers from poor clarity in some parts, mainly due to the repetition of concepts already discussed elsewhere in the text (sometimes even within the same section), which make the text confusing and prone to redundant information, or even due to inclusion of content in sections which are not the most appropriate in that case. This obscures the research value, confusing the reader and making the text unnecessarily long. In some cases, responses you gave to editors were clearer and more concise than what you wrote in the paper. In some others, information you provided in the letter is missing from the revised paper. I would recommend to integrate these pieces of information from the response letter into the paper, carefully avoiding repetitions.

---

## Round 0.3 · accepted · Accept

The reviewers appreciated the changes made and therefore I can recommend this article for acceptance.

·

Basic reporting

Thank you for your thorough answers and also your modifications of according paragraphs in your re-submitted manuscript.

Experimental design

No comment.

Validity of the findings

No comment.

·

Basic reporting

The “Introduction” session provides adequate context and makes clear what the motivation is. Literature is well referenced, paraphrased, and relevant, and is presented in the introduction session. English was improved with respect to previous manuscript submission.

Experimental design

Suggestions to improve the previous manuscript version were considered and implemented.

Validity of the findings

General requirements from the previous review process were satisfied